# Confocal Laser Microscopy in Neurosurgery: State of the Art of Actual Clinical Applications

**DOI:** 10.3390/jcm10092035

**Published:** 2021-05-10

**Authors:** Francesco Restelli, Bianca Pollo, Ignazio Gaspare Vetrano, Samuele Cabras, Morgan Broggi, Marco Schiariti, Jacopo Falco, Camilla de Laurentis, Gabriella Raccuia, Paolo Ferroli, Francesco Acerbi

**Affiliations:** 1Department of Neurosurgery, Fondazione IRCCS Istituto Neurologico Carlo Besta, 20133 Milan, Italy; francesco.restelli91@gmail.com (F.R.); ignazio.vetrano@istituto-besta.it (I.G.V.); samuele.cabras.ferrarinii@gmail.com (S.C.); morgan.broggi@istituto-besta.it (M.B.); marco.schiariti@istituto-besta.it (M.S.); jacopo.falco910@gmail.com (J.F.); camilla.delaur@gmail.com (C.d.L.); gabriellaracc@gmail.com (G.R.); paolo.ferroli@istituto-besta.it (P.F.); 2Neuropathology Unit, Fondazione IRCCS Istituto Neurologico Carlo Besta, 20133 Milan, Italy; bianca.pollo@istituto-besta.it

**Keywords:** confocal laser endomicroscopy, confocal laser microscopy, brain tumors, fluorescein

## Abstract

Achievement of complete resections is of utmost importance in brain tumor surgery, due to the established correlation among extent of resection and postoperative survival. Various tools have recently been included in current clinical practice aiming to more complete resections, such as neuronavigation and fluorescent-aided techniques, histopathological analysis still remains the gold-standard for diagnosis, with frozen section as the most used, rapid and precise intraoperative histopathological method that permits an intraoperative differential diagnosis. Unfortunately, due to the various limitations linked to this technique, it is still unsatisfactorily for obtaining real-time intraoperative diagnosis. Confocal laser technology has been recently suggested as a promising method to obtain near real-time intraoperative histological data in neurosurgery, due to its established use in other non-neurosurgical fields. Still far to be widely implemented in current neurosurgical clinical practice, this technology was initially studied in preclinical experiences confirming its utility in identifying brain tumors, microvasculature and tumor margins. Hence, ex vivo and in vivo clinical studies evaluated the possibility with this technology of identifying and classifying brain neoplasms, discerning between normal and pathologic tissue, showing very promising results. This systematic review has the main objective of presenting a state-of-the-art summary on actual clinical applications of confocal laser imaging in neurosurgical practice.

## 1. Introduction

Despite the most recent therapeutic advancements, prognosis of brain tumors still remains poor [1,2]. Surgical resection represents a major component in the standard of care for the treatment of brain cancers. Various clinical studies have demonstrated that extent of resection (EOR) correlates with improved outcomes, especially in conjunction with adjuvant therapies such as radio-chemotherapy [3,4]. Nonetheless, achieving a complete tumor removal is not always feasible, since distinction between normal and pathological tissue is often difficult, especially when approaching tumor margins [5]. 

Through the last decades, a number of tools and devices have been used and studied to improve EOR, such as intraoperative ultrasound, neuronavigation and fluorophores that may enhance tumor tissue visualization during surgery [6,7,8]. The implementation of these techniques demonstrated to improve identification of tumor margins, leading to more extensive resections. However, by date, histopathological techniques are the only weapons that may provide microscopic identification of tumor cells and effective infiltration at tumor margins. While histopathological analysis still remains the gold-standard for diagnosis, frozen section is nowadays the most used, rapid and precise intraoperative histopathological method that permits an intraoperative differential diagnosis. Unfortunately, the results obtained using this technique can be misleading or nondiagnostic, particularly in cases of mechanical tissue disruption from the resection process [9,10]. Moreover, this method owes other significant drawbacks, requiring long time to analyze tissue samples (often 20–30 min), needing to be processed and analyzed outside the operating room (OR). Effective efficacy of this technology to reveal the exact diagnosis is questioned as well, in fact a diagnostic discrepancy between frozen and permanent sections is reported to be as high as 2.7% in intracranial pathologies analysis [10]. Such diagnostic unpredictability is further complicated by the inherent heterogeneity of brain tumors. For instance, gliomas can contain high-grade populations nested within a low-grade stroma, representing a significant challenge for the pathologist. All these aspects contribute to render frozen sections still unsatisfactory to reveal the histological features necessary for the final diagnosis for a possible guidance for intraoperative decision regarding EOR [9,10,11].

In this context, confocal laser technology is an imaging technique that provides microscopic information of tissue in real time. Such technique has already been integrated into the current clinical practice in non-neurosurgical fields. For instance, confocal laser endomicroscopy (CLE) has been studied with very promising results in tissue selection process for biopsy procedures in general surgery, where it can limit the number of samples needed for diagnosis, or during removal of lesions for whom a judicious study of pathological margins is mandatory in the fields of gastroenterology, urology and gynecology [12,13]. Still far to be routinely used in neurosurgery, in recent years CLE has been proposed also in this field. The first studies in mouse glioblastoma (GBM) models were focused on the ability to distinguish normal brain, microvasculature and tumor margins [14,15,16]. Later on, after the first preclinical experiences, feasibility of CLE in human brain tumor surgery was questioned and studied through both ex vivo and in vivo experiences with promising results [11,17,18,19]. Taken together, neurosurgical literature suggests that this technology is capable of providing multiple optical biopsies intraoperatively with histological imaging of the cells at a microscopic level, representing the first technique able to provide in vivo histopathological data from fresh tissues with an “on-the-fly” methodology. Exact sensibility, specificity and accuracy in identifying tumor cells and the real role that this technology may have in the near future in neurosurgery is still under deep investigation.

Main objective of this systematic review is to present an update on actual clinical applications of confocal laser imaging technology in neurosurgery.

## 2. Materials and Methods

### 2.1. Literature Search and Screening Process

A comprehensive, systematic literature search was performed in January 2021. MEDLINE (PubMed), EMBASE and SCOPUS were searched using the following search strings in the “Title/abstract” field: “confocal AND neurosurgery”, “confocal AND glioma”, “confocal AND brain tumor”, “endomicroscopy AND neurosurgery”, ”endomicroscopy AND brain tumor”, “endomicroscopy AND glioma”, “confocal imaging AND glioma”, “confocal imaging AND brain tumor”, “confocal imaging AND neurosurgery”, “confocal endomicroscopy AND glioma”, “confocal endomicroscopy AND brain tumor”, “confocal endomicroscopy AND neurosurgery”, “Cellvizio AND glioma”, “Cellvizio AND brain tumor”, “Cellvizio AND neurosurgery”, “Endomag1 AND glioma”, “Endomag1 AND brain tumor”, “Endomag1 AND neurosurgery”, “Convivo AND glioma”, “Convivo AND brain tumor” and “Convivo AND neurosurgery”.

Search was limited to articles in English, without publication time limits (published article until 31 January 2021). All titles and abstracts were checked by two different researchers (F.R. and F.A.). Duplicates were removed and all relevant articles were collected and studied. Relative bibliographies were hand-searched to identify further relevant literature. If there was a difference in opinion on suitability of the works among the researchers, a consensus was reached by consulting a third senior reviewer (I.V.). To note, in this first-phase pure reviews on the topic were not excluded a priori, to prevent any loss of data and to further broaden the search process for studies that might have been missed through the first search. Due to the large differences in methodologies and patients’ cohort available in different studies, the literature search did not strictly follow the criteria for a systematic review. Anyway, we tried to identify the highest quality of available evidence for each specific theme.

### 2.2. Eligibility Criteria

After the screening process, remaining articles were analyzed full-text by two authors (F.R. and F.A.) to check their relevance and eventual accordance to the below-mentioned inclusion and exclusion criteria. Briefly, only clinical studies concerning in vivo or ex vivo applications of confocal imaging technologies in neurosurgery were analyzed. The clinical results of works with both a preclinical and a clinical experimental part were included as well. The following inclusion/exclusion criteria were followed:

Inclusion criteria:-Clinical works focused on confocal imaging technology application in neurosurgery.-Preclinical works with a subsection on clinical application of confocal imaging technology in neurosurgery.

Exclusion criteria:-Correspondences.-Comments.-Letters to the editor.-Proceedings/conference papers.-Case reports.-Reviews.-Purely preclinical studies.

### 2.3. Data Extraction

All the included studies were analyzed and following data were extracted and summarized in tables using Microsoft Excel (version 365, Microsoft Corporation, Redmond, WA, USA) and Mendeley Desktop for Mac. Authors, year of publication, confocal imaging technology used, type of fluorescent dye implemented (if used), type and number of tumors treated, main findings, blinding grade of the study were collected and reported by two authors (F.R. and F.A.) after recruitment of all eligible studies. Considering the extreme heterogeneity of the studies available and the limited number of published works, we present the data as a comprehensive (narrative) review.

### 2.4. Statistical Analysis

Statistical analysis of the data, for the purpose of a metanalysis, was not possible due to substantial heterogeneity in study design and populations.

## 3. Results

A total of 1495 hits were recorded by the first search among the three databases (PubMed 253, EMBASE 399, SCOPUS 843). Of such works, 814 were screened reading titles and abstracts, removing duplicates. Finally, 112 full-text articles were assessed for eligibility, finding 33 articles suitable for the final analysis (Figure 1).

### 3.1. In Vivo Experiences

Available studies reporting in vivo experiences are reassumed in Table 1 (7 works).

The first authors to study the applicability of confocal imaging in vivo in neurosurgery was the group of Sanai and colleagues in 2011 [20]. Sodium fluorescein (SF) was used as contrast enhancer and in vivo images were obtained in 31 brain tumors (25 intra-axial, 6 extra-axial) using an Optiscan 5.1 system, comparing the confocal captions with common histopathological sections. For the first time in vivo confocal hallmarks of different brain tumors such as high-grade gliomas (HGGs, vascular proliferation, dense cellularity and irregular cellular phenotypes), low-grade gliomas (LGGs) and meningiomas were reported, appearing at least comparable to the ones found on common pathological examination. Subsequently, in 2011 the same group analyzed a subgroup of 10 patients who underwent resection of LGGs at their center [21]. As expected, no 5-aminolevulinic acid (5-ALA) signal was found with Pentero microscope, whereas CLE revealed sparse intratumoral fluorescent spots that after histological biopsy were confirmed as tumor infiltration areas. To note, the authors reported an unsatisfactorily view of cell contours, if compared to SF administration. In 2012 a subsequent study analyzing 50 patients affected by different central nervous system (CNS) tumors was carried out by Eschbacher and colleagues [22]. The authors studied 88 biopsy locations in vivo, obtaining multiple images for each location and sending biopsies to common histological analysis, founding a surprisingly high correspondence between CLE and histological findings, including the identification of many pathognomonic cytoarchitectural features of various brain tumors. Moreover, in this study 28 selected images were presented to a blinded neuropathologist, without prior experience in CLE interpretation, reporting a 92.9% accuracy in making diagnosis (as resulted by the ex vivo analysis). In 2016 Martirosyan reported a study where CLE imaging (Optiscan 5.1) was used in the resection of intracranial neoplasms in 74 consecutive patients with intraoperative in vivo and ex vivo CLE imaging after intravenous injection of SF, comparing them to frozen sections and histological sections analysis [11]. Locations of imaging included both normal brain and regions of obvious tumor, with tumor margins as well. For the first time a real report of sensibility/specificity for this technology in vivo was performed. In vivo, CLE images were diagnostic for 45.98% and CLE specificity and sensitivity were, respectively, 94% and 91% for gliomas and 93% and 97% for meningiomas (comparable to sensibility/specificity of frozen sections in respect to definitive histological diagnosis). This article reported also procedural and practical aspects linked to in vivo CLE application (mean time for 4 optical biopsies per case of 6 min). In the same year, another group headed by Pavlov studied another CLE system, the Cellvizio, in an in vivo setting [17]. The authors studied the feasibility in humans in a series of 9 brain neoplasms surgeries with both 5-ALA and SF administration. The authors reported autofluorescent spots when imaging normal brain before resection, attributed to lipofuscin accumulation in neurons. Overall, Cellvizio images enabled differentiation of healthy “normal” tissue from pathological tissue during open surgeries and stereotactic biopsies using SF, although it was impossible to distinguish among different grades of gliomas. 5-ALA confocal patterns were difficult to be determined, and this was mainly due to suboptimal excitation laser wavelength. Since these initial in vivo experiences, research have slowed mainly due to the lack of appropriate sterile sheaths for CLE machines, that rendered difficult sterilization processes on a large scale. In 2019 Charalampaki and colleagues demonstrated the feasibility of using in vivo during the same surgical procedure a multispectral fluorescence microscope, able to merge white light and ICG signal, and CLE probe (Cellvizio) [23]. The authors performed optical biopsies in 13 cases of brain neoplasms in the fluorescent part of the tumor, in transitional zones and in normal tissue. Even though no data on sensibility and specificity of such technique were reported and blindness grade was unspecified, the use of the CLE tool was considered important given that they were able to achieve both an improved representation of the borders between tumor (multi-color highlight) and normal tissue as well as a significant improvement in the representation of immediate histological diagnosis compared with common hematoxylin and eosin (H-E) staining. In 2020 an appropriate sterile sheath became available for the Convivo system by Zeiss (new generation system, with an increased lateral resolution of about 0.5 µm and Z-axis resolution of about 4.5 µm and an automatic Z-stack acquisition function). Convivo system was used in the last in vivo work available in the literature, by Hohne and colleagues [24]. The authors evaluated the handling, operative workflow and visualization of Convivo system in 12 cases of different CNS tumors. In particular, three different imaging positions in relation to the tumor were chosen: the tumor border, tumor center and perilesional zone. Respective diagnostic sampling with H-E staining and matching intraoperative neuronavigation and microscope images were provided, concluding that such system could be used safely in vivo, allowing excellent visualization of microstructures in the surgical field. No sensibility/specificity or procedural calculations were reported in this study.

An in vivo case operated on at our Institute is reported in Figure 2.

### 3.2. Ex Vivo Experiences

All the clinical studies focused on ex vivo application of confocal imaging in brain tumor surgery are listed in Table 2 (21 works).

The first study to address the use of CLE in brain tumor surgery was the one of Schlosser and colleagues in 2010 [25]. Analyzing CLE imaging on ex vivo biopsies from 9 HGGs and 3 meningioma cases, the authors showed for the first time the promising role that CLE may have in brain tumor surgery, demonstrating ex vivo CLE characteristics of glioma tissue (such as necrosis, hypercellularity and mitoses) and raising the possibility to obtain targeted biopsies, which could increase the reliability of the diagnosis when multiple cell types contribute to a tumor, reducing sampling errors. To note, as also stated by the authors in their work, one of the limitations of this first study was the utilization of topical acriflavine (AF) as contrast enhancer. Given that this dye is not suitable for in vivo brain tissue study (toxic concerns, see below), the authors suggested the use of SF for the next studies, given also its proven neuro-oncological significance in neurosurgery [6]. Anyway, following such proof-of-principle study, other authors reported novel experiences in ex vivo brain tumor surgery with topical AF administration. In 2012 Foersch and colleagues reported the use of ex vivo AF or SF and CLE imaging in 15 brain tumors [26]. Tumor specimens and healthy brain tissue were collected from patients who underwent surgery and then colored topically with either AF or SF. The authors reported many of the common CLE characteristics of various kind of CNS tumors such as GBM (rampant tumor growth, atypical nuclear-to-cytoplasm ratio, mitoses), meningiomas (psammoma bodies) and epidermoid tumors (pentagonal cells of epithelial origin, mosaic pattern) and calculated for the first time an accuracy in identifying a correct diagnosis using CLE imaging. A total number of 175 representative endomicroscopic images of 7 different human tissues were evaluated by 5 different raters blinded to macroscopic appearance and histology of the different tumors. Detection rate was 87.14% for not clinical experts, whereas clinical experts gave correct diagnosis in 93.33% of the cases. Wirth and colleagues and then Snuderl and colleagues in 2012 and 2013, respectively [27,28], showed the results of ex vivo imaging of brain tumors using a novel multimodal CLE system, functioning on both reflectance and fluorescence mode after staining of surgical specimens with methylene blue (MB). In these two studies reflectance and fluorescence images provided distinctly different and complementary information regarding brain tissue morphology (fluorescence images visualized different types of cells, including neurons and glial cells, vascular proliferations and necrosis, while reflectance images showed better cell morphology features, such as membranes and contours). In the 2013 work, moreover, analyzing a subset of images from 10 preselected cases, the authors found a sensitivity/specificity in identifying normal vs abnormal brain and in differentiating among glial vs non-glial neoplasms of 95%/100% and 83%/90%, respectively. Neuropathologists were able to give the final correct diagnosis with a sensibility/specificity of 88%/100% (*p* = 0.02 and *p* < 0.0001, respectively) [28]. The group of Wirth and colleagues went further in 2015, analyzing 14 cases of CNS tumors, confirming that optical images of tissues stained with Demeclocycline (DMN) display good correlation with histopathology, suggesting a possible role for such dye in in vivo imaging, as this antibiotic may be administered to patients orally [29]. Breuskin and colleagues published two works regarding CLE imaging in 2013 and 2017, using red laser EndoMag1 on more than 50 brain tumors specimens in the first work. Diagnostic accuracy was calculated in the subsequent work as well, analyzing 100 tissue biopsies from 100 tumor cases, reporting a diagnostic sensitivity of 81% for HGGs, 82% for LGGs and 95% for meningiomas [19,30].

In 2014 Georges and colleagues imaged 2 fresh human brain tumor biopsy specimens: 1 affected by radiation necrosis and 1 with known GBM tissue, suggesting and proposing CLE for the first time as a valuable method for differentiating among tumor tissue and radiation necrosis of surgical specimens prior to biobanking purposes [31]. In 2017 the same group used confocal reflectance microscopy, without administration of fluorescent dyes, to rapidly identify histopathological features from fresh human brain tumor biopsies [32]. A neuropathologist and surgical pathologist masked to the imaging results evaluated independently a subset of images. In these evaluations, 100% of images reviewed by the neuropathologist and 95.7% of images reviewed by the surgical pathologist were correctly diagnosed as lesional or non-lesional. Furthermore, 97.9% and 91.5% of cases were correctly diagnosed as tumor or not tumor by the neuropathologist and surgical pathologist, respectively, while 95.8% and 85.1% were identified with the correct diagnosis.

In 2015 Charalampaki presented the results of 150 cases operated on with the aid of CLE using topical AF as contrast enhancer, reporting specific descriptive characteristics of the tumors investigated [18]. A year after, the same group presented the results of 258 patients, in this case calculating an accuracy for CLE imaging as well. Preselecting images from 258 tissue samples the authors calculated an overall accuracy of 89% [33]. In the same year the possible implementation of acridine orange (AO) as contrast enhancer in ex vivo CLE imaging was tested by Forest and colleagues, analyzing on 20 consecutive surgical biopsies/tumor excisions obtained in 19 patients (7 infiltrating gliomas, 9 meningiomas and 3 metastases) using a confocal microscope specifically designed for dermatological pathologies [34]. In this study such system was proposed as a viable method to quickly identify tumor tissue without tissue loss, differentiating tumors and assessing most of grading criteria.

In 2017, following the promising results of Sanai and colleagues on the implementation of 5-ALA in in vivo confocal imaging (see relative section), Wei and colleagues developed a handheld video- rate optical-sectioning microscope optimized to visualize and quantify subcellular protoporphirin IX (PpIX) expression [35]. This technology seemed to be promising for increasing PpIX detection in gliomas, improving performances reported by Sanai years before. Following 5-ALA philosophy, in 2017 and 2019 the group of Yoneyama studied the possibility of using fluorescence intensity and a detailed bright-spot analysis to increase 5-ALA detection, thus aiming to detection of tumor boundaries and tumor infiltration at borders [36,37]. In particular, from the fluorescent intensity of the image pixels of confocal imaging, a histogram of pixel number with the same fluorescent intensity was obtained. The fluorescent bright spot sizes and total number were compared between the marginal and normal regions, finding that fluorescence intensity distribution and average intensity in the tumor were different from those in the normal regions; moreover, the bright-spot size and number in the infiltrating tumor were different from those in the normal region, suggesting that these two mathematical calculations may help in identifying tumor from non-tumor areas using 5-ALA as contrast enhancer. The same bright-spot analysis was conducted in 2019 on GBM with and without 5-ALA-induced fluorescence as well as for LGGs and other brain tumor types, suggesting the method as potentially useful also for tumors with no 5-ALA-derived red fluorescence and other nervous system tumors [37].Martirosyan and colleagues in 2018 reported the results of ex vivo analysis of 106 cases of brain tumors, marked with various fluorescent dyes (AO, cresyl violet (CV), indocyanine green (ICG), MB), comparing qualitative results and calculating diagnostic accuracy. Great attention was made in quality of images and in describing confocal characteristics of different brain tumors under CLE visualization [38]. As anticipated, an overall 89% of gliomas, 87% of meningiomas and all pituitary adenomas were diagnosed correctly based on CLE images by neuropathologists. AO and AF were found to be the best topical contrast enhancer for the great majority of tumors studied and investigated.

The group of Mooney in 2018 focused on pituitary adenomas analysis. In their work biopsy specimens from 11 patients with suspected pituitary adenomas were studied under confocal reflectance microscopy and for all specimens, confocal contrasted cellularity, tissue architecture, nuclear pleomorphism, vascularity and stroma. Blinded interpretation of preselected images resulted in a 94% accuracy in making correct diagnosis [39].

More recently, Belykh and colleagues reported various works concerning ex vivo CLE imaging. In 2018 the authors performed CLE imaging, Z-stack acquisition and 3D image rendering of 31 human tumors, including meningiomas, gliomas and pituitary adenomas. For the first time in this work Convivo system by Zeiss, a second-generation CLE system, was used [40]. Differently from old generation systems, such machine presented some specific novel features such as an increased lateral resolution of about 0.5 µm, a Z-axis resolution of about 4.5 µm and an automatic Z-stack acquisition function. In this work the authors provided 3D images of different kind of brain tumors, suggesting that this technology may afford an increased spatial understanding of tumor cellular architecture and visualization of related structures compared with two-dimensional images. In 2020, the same group tested for the first time the feasibility of positioning the Convivo probe in a trans-sphenoidal corridor in cadaver heads and obtained 19 biopsies from nine patients who underwent pituitary adenoma surgery for ex vivo imaging at various times after fluorescein injection [41]. A blinded board-certified neuropathologist analyzed the images, resulting in a 8/13 pituitary adenomas correctly identified through CLE analysis using frozen sections as the standard. In another work the authors used Convivo on 47 patients with 122 biopsies (29 HGGs). In this work, a highly detailed sensibility/specificity study was performed, disclosing a positive predictive value of CLE optical biopsies of 97% for all specimens and 98% for gliomas, a specificity of CLE of 90% for all specimens and 94% for gliomas. To note, the author described an improved image quality and increased percentage of accurately diagnosed images from 67% to 93% when a second SF injection was performed (after a mean of 2.6 h after the first injection). The last available work to date concerning ex vivo imaging in brain tumors surgery was the work of our group in 2020. We prospectively studied ex vivo the ability of Convivo to confer an intraoperative first-diagnosis during GBM removal, by blindly comparing intraoperative CLE and frozen/permanent sections results at both central core and tumor margins of tumors [42]. We were interested in both identifying the ability of such system in offering an intraoperative diagnosis, along with the ability to categorize morphological patterns, such as cellularity, vascularization and necrosis. Blindly comparing Convivo and frozen sections images we obtained a high rate of concordance in both providing a correct diagnosis and categorizing patterns at tumor central core (80% and 93.3%, respectively) and at tumor margins (80% for both objectives) with lower rates if compared to permanent sections (total/partial concordance in 80% and 86.7% for diagnosis and morphological categorization, respectively, with lower results at tumor margins).

An ex vivo case operated on at our Institute is reported in Figure 3.

### 3.3. Contrast Enhancers Used during In Vivo and Ex Vivo CLE Imaging

To date, various fluorescent dyes with different staining characteristics have been used in clinical settings. Among them, SF is one of the most used, thanks to its proven oncological effect and its possibility to be used in vivo. From a pharmacological viewpoint, the contrast agent quickly diffuses across capillaries, highlighting first blood vessels (similarly to an ICG injection), then permeating interstitial spaces for up to 30 min. Adverse events such as acute hypotension or anaphylaxis were rarely described [43,44]. As contrast enhancer for CLE imaging this dye has been extensively studied by the Barrow Neurological Institute group since 2012 in both in vivo and ex vivo studies, confirming the feasibility and utility of it as confocal imaging enhancer [11,22]. In a recent works by Belykh and colleagues, the dye was studied for ex-vivo pituitary tumors analysis, identifying the correct timing for SF administration in respect to biopsy procedure [45]. In 2020 the authors reported for the first time the increasing diagnostic power that a second intravenous injection of such dye may confer in the analysis of intracranial lesions [41]. Thanks to its proven effect in increasing EOR and ability to enhance CLE images, highlighting tumor cells in fluorescent areas, to date this dye appears to be one of the best contrast enhancers that can be coupled to confocal imaging. 5-ALA, instead, was firstly used by Sanai and colleagues in an in vivo experience in 2011 using Optiscan 5.1 system [20]. The dye permitted to identify infiltration areas under CLE imaging in LGGs. Pavlov in 2016 evaluated Cellvizio’s ability to excite and detect 5-ALA intraoperative fluorescence in 3 patients with HGGs, founding sensitivity of Cellvizio utilizing 5-ALA as still insufficient [17]. Years later, Wei and colleagues developed a handheld video- rate optical-sectioning microscope optimized for visualizing and quantifying subcellular PpIX expression, improving the possibility to detect intracellular PpIX [35]. Similar results in terms of improving the interpretation of 5-ALA induced fluorescence were reported in the same years by Yoneyama and colleagues, that set up a detailed fluorescence intensity analysis algorithm along with a bright-spot analysis (using the relative intensity of signal of each pixel of confocal images obtained) to distinguish among tumor and non-tumor regions. Bright-spot analysis was also useful in GBM not characterized by 5-ALA induced fluorescence and in other tumor types [37]. Unfortunately, real performance and sensibility/specificity results using such technology are still awaited.

Intracellular and nuclear staining can be achieved with topical contrast agents. They are easily applied and they do not regularly carry the risk of systemic side effects but, to date, no dye is officially licensed for in vivo use in humans. Given that such dyes typically do not penetrate deeply, they are routinely used on ex vivo analyses and avoided in in vivo ones. The most common agent is AF, which stains the nuclei and has been applied by different authors [25,26]. When stained with AF, satisfactory contrast for endomicroscopic imaging was described by various authors for different kind of tumors such as HGGs and meningiomas [25,26]. When fluorescein and topical AF are combined, it is possible to calculate nuclear cytoplasmic ratios, which are useful indicators for cellular differentiation. Even though different authors demonstrated an anti-tumor activity of AF in cell culture and a positive effect in slowing tumor progression in preclinical models [46,47], there is still a considerable concern about a potential mutagenic effect of this dye, so it has limited use in in vivo experiments [26].

Other topical dyes (or precursors) that have been used for confocal ex vivo imaging are CV, AO or MB. MB is an in vivo as well as in vitro staining agent that is safe to use and without toxic nature for the patient. In histology, it stains nuclei, making their examination favorable [30]. Apart from this histological use, MB has been used a spray dye in gastroenterological endoscopic procedures in order to visualize altered tissue; such dye is FDA approved for in vivo use. Looking at neurosurgical field, various groups studied the utility of this dye in ex vivo confocal imaging, demonstrating that reflectance images provide information about morphology and vascularity of the specimens, complementary to that provided by fluorescence images after MB staining [27]. In particular, with MB nucleus staining mimics the staining pattern of hematoxylin while reflectance images mimic the staining pattern of eosin [28]. However, various clinical observations suggest the possibility of adverse neurologic effects after MB injection in the CNS, reason for which other groups analyzed the utility of another agent for identifying tumor cells ex vivo: DMN. As a matter of fact, DMN represents a derivative of tetracycline approved for in vivo use. DMN has been shown to demarcate tumors in human skin tissue [48], and the group of Wirth in 2014 and then in 2015 analyzed specific imaging characteristics of DMN-labeled HGGs tissue. The authors found high contrast enhancement characteristics and a similar staining pattern as observed in conventional H-E histopathologic preparations of GBM, metastases, meningiomas and pituitary adenomas [29,49].

AO was instead studied by Forest and colleagues in a 2015 work [34]. The authors confirmed ex vivo confocal microscopy imaging with this dye as a viable method to quickly identify tumor tissue without tissue loss, differentiating tumors and assessing most of grading criteria. The same dye was used by Martirosyan and colleagues in a 2018 work where AO demonstrated to act similarly to AF [38]. While AF and AO produced high-quality staining of extracellular and intracellular structures (predominantly nuclei and cellular membranes), AF had a propensity to label more extracellular structures than AO. In summary, both AF and AO provided good quality fluorescence in all cases, except for cavernous malformations.

CV was used by Martirosyan and colleagues to stain brain tumors specimens [38]. Unlike AF and AO, that stained nuclei primarily, CV caused diffuse fluorescence of the cytoplasm that highlighted large diffuse areas of the different images. Cell morphology was defined by fluorescent cytoplasm encased within the cell membrane while intracellular components were identified as dark shadows over the fluorescent cytoplasm. In the same work the authors also investigated the role of ICG as contrast enhancer for brain neoplasms. While used in various preclinical works [50,51], just few studies studied the role of this fluorescent dye in confocal imaging, exhibiting minimal fluorescence; no meaningful histopathological patterns were identified using ICG in the work of Martirosyan in 2018 [38]. Charalampaki in 2019 studied the possibility of using both multispectral fluorescence microscope and CLE (Cellvizio) in vivo during the same surgical procedure, using ICG as contrast enhancer [23]. The authors found the best results for the multispectral infrared imaging of meningiomas, neurinomas and metastatic tumors, while HGGs and LGGs did not uptake ICG efficiently; nevertheless, confocal imaging was able to detect at a cellular level such tumors that were not highlighted with the fluorescent surgical microscope.

### 3.4. Summary on Sensibility, Specificity and Diagnostic Accuracy of Confocal Imaging Technology in Brain Tumors Surgery

To date, analyzing available and pertinent literature, there is insufficient data to make any definitive conclusion on the real usefulness of confocal imaging technology in improving intraoperative tumor diagnosis and eventually extent of resection. Overall, all studies (both benchtop confocal systems and portable confocal systems experiences) have described the methodological feasibility of obtaining a diagnosis and categorizing characteristics of different brain tumors in ex vivo and in vivo studies. As a matter of fact, many in vivo and ex vivo experiences tried to draw some conclusions on the topic calculating different quantitative parameters linked to the diagnostic power that this technology may have.

It has to be said that, from a methodological point of view, a real sensibility and specificity calculation would be interesting especially on biopsies taken at tumor border (to possibly identify tumor tissue and increase extent of resection) and would be possible only for pure in vivo experiences, given the need of optical biopsies on healthy brain parenchyma (negatives). Nevertheless, many authors included “negatives” also in ex vivo studies, for instance considering specimens obtained in areas of not tumoral “reactive tissue” near to tumor border as true negatives. Moreover, the great majority of available studies is biased by: (1) low number of biopsies per case (the recent availability of probes that can be used directly in vivo in OR hopefully will improve such aspect); (2) low number of biopsies performed at tumor border; (3) unspecified or “low controlled” blindness of the study from the pathologist point of view. Hence, considering ex vivo experiences, many studies have demonstrated different sensibility/specificity for HGGs, often over 80/85% and also higher for extra-axial tumors [28,32,41]. In particular, Snuderl and colleagues in 2012 analyzed ten preselected images from tumor cases, blinding pathologists to histological results, simulating a frozen section analysis, founding that identification of grade and tumor type by trained neuropathologists resulted to have a sensibility of 88% and specificity of 100% [28]. Fewer considerations may be drawn for looking at other tumor types, such as LGGs, schwannomas or pituitary adenomas, that were studied in less works [21,39,45,52]. Considering in vivo experiences, to date only one study reported a solid statistic compartment, showing sensibility and specificity calculation for brain tumors in vivo, demonstrating a sensibility/specificity for HGGs of 91% and 94%, respectively and 97%/93% for meningiomas, respectively [11]. Further studies in this field are awaited to discover how sensitive and specific such system may be when used to check for tumor tissue at borders.

### 3.5. Descriptive Confocal Imaging Patterns of Normal Brain and Different Brain Tumors

#### 3.5.1. Normal Brain

Snuderl and Wirth demonstrated confocal reflectance and fluorescence microscopy images of normal brain by ex vivo specimens using a custom-made confocal benchtop microscope, using MB as fluorescent labeling for cells [27,28,29]. Reflectance images of both white and gray matter demonstrated lattice-like architecture of normal brain, with well-organized processes. Myelin produced a dense, white, mesh-like appearance. In particular, given the presence of the myelin on axons, the reflectance images of white matter resulted brighter than gray matter [27]. The fluorescence image of normal gray matter presented well-defined neuronal cell bodies while a lower identification of white matter due to the dielectric properties of myelin preventing axons to uptake MB. Similar findings were reported by the group of Charalampaki, using Cellvizio as CLE and AF 0.05% for topical staining [18,33]. In the 2020 study of Belykh, for the first time three types of autofluorescence, mainly correlated to normal tissue, were distinguished and classified: scant, patchy and dense. Scant autofluorescence was frequently observed in non-glioma samples that were deemed nondiagnostic in this study [41].

#### 3.5.2. HGGs

Neovascularization, dense cellularity and irregular cellular phenotypes are characteristic hallmarks of HGGs on confocal imaging, easily seen on both ex vivo and in vivo specimens [25,26]. When SF is administered as a contrast enhancer, images are typically dominated by extravascular SF concentrated in clusters and nests around cellular structures showing hypercellularity and pleomorphism. Necrosis, when present, is evident as waves of varied cellular density [20]. As noted by Eschbacher, occasionally tumors cells may be distributed in a diffuse myxoid matrix on histological staining: such aspect may affect the permeability of the fluorescein, reducing CLE images quality [22]. Using confocal reflectance and fluorescence imaging with MB as a cell-labeler, Wirth and Snuderl showed the abundance of tumor cells with lose of normal cytoarchitecture with both confocal reflectance and fluorescent imaging [28,49]; necrotic areas were characterized by the loss of signal in both kind of modalities. Similar results were reported by the studies of Breuskin and colleagues [19,30]. Martirosyan and colleagues in 2018, analyzing the efficacy of different topical fluorescent dyes, reported gliomas as best visualized with AF and AO staining [38].

Anaplastic oligodendroglioma and gliosarcoma features were described by Sanai and Eschbacher [20,22]. In oligodendroglioma cellular atypia was usually evident on confocal images. Morphologically, the neoplastic cells consisted of irregular gray to dark gray cell bodies; neuronal satellitosis may also be seen. The gliosarcoma usually demonstrated fascicles of markedly atypically elongated tumor cells, which appeared dark against a bright fluorescent background.

#### 3.5.3. LGGs

Morphologically, tumor cells of astrocytic origin demonstrate increased pleomorphism compared with those of oligodendroglial origin in various works, but less cellular and more diffuse than oligodendrogliomas [22]. In particular, astrocytic cell bodies usually appear more elongated than neoplastic oligodendrocytes. In a work by Sanai and colleagues, neovascular proliferation was usually less evident if compared to higher grades, distinctions in cell density and cellular morphology corresponded with T2-weighted signal abnormalities on MR imaging [20]. Using MB as cell dye, Snuderl and Wirth in 2012 and 2013 studied LGGs ex vivo using a custom-built benchtop confocal microscope [28,29,49]. The fluorescence images appeared to be a good indicator of abnormally increased cellularity.

#### 3.5.4. Ependimomas and Subependimomas

In the work of Eschbacer in 2012 ependymoma demonstrated discrete clusters of dark cell bodies around vessels bearing bright intraluminal SF [22]. Acellular zones were visible between cell bodies and the vessel wall, consistent with corresponding processes viewed on H-E. Other areas demonstrated ribbons of nuclei adjacent to acellular regions, possibly representing pseudorosettes. Subependimomas were deeply studied in a work by Martirosyan and colleagues (2018) with the aid of AF and AO staining. Clusters of isomorphic nuclei embedded in a dense, fine, glial fibrillary background were the characteristics associated with this tumor subtype [38].

#### 3.5.5. Meningiomas

Classic meningothelial meningiomas are characterized by largely uniform tumor cells organized as dense sheets of cells without evidence of whorls or psammoma bodies (characterized as spherical, a-nuclear, whorled structures on both confocal and H-E sections) that can be seen in both cases of contrast agents used as contrast enhancers or labelers [19,26,28,29,53]. In contrast, fibrous meningiomas may contain spindle-shaped cells, easily distinguishable from adjacent normal parenchyma [20]. To note, as opposed to malignant tumors such as GBM, nuclear shape is usually more uniform and typically well-ordered [26]. Sanai and colleagues reported difficulty in discriminating between grade I and II meningiomas, although at confocal imaging in higher grade lesions in vivo tumor borders were much poorly defined and more atypical cellular patterns were often seen [11,20]. Occasionally, tumors showed cells with intracellular round-to-oval clear centers, corresponding with nuclear pseudo-inclusions on H-E sections.

In addition, using cell labeling such as MB the classical typical meningothelial whorls can be seen in both reflectance and fluorescence confocal images [28,29]. Using AO, AF, MB, CV and ICG as contrast enhancers, Martirosyan and colleagues found meningiomas as better identifiable with AF and AO staining [38].

#### 3.5.6. Schwannomas

Schwannomas usually have much larger streaks of fibers, with cells less prominent than in meningiomas, although presenting a fibrous aspect [19,30]. In many works, fascicles of cells with elongated cytoplasmic processes appeared to correlate with Antoni A regions on H-E sections [22]. Foersch demonstrated clusters of atypically shaped nuclei in schwannomas cases, that most likely represented Verocay bodies [26]. Neither H&E sections nor confocal images exhibited necrosis, mitotic figures or significant cytological atypia for any of the schwannomas of the work of Eschbacher in 2012. AF and AO were the best fluorescent dyes to study these tumors as reported by Martirosyan in 2018 [38].

#### 3.5.7. Craniopharyngiomas

Imaging of craniopharyngiomas revealed structures consistent with wet keratin admixed with nests of cells with epithelial features in the work of Martirosyan [11]. Foersch in 2012 identified nuclear morphology of craniopharyngiomas ex vivo as elliptic shapes, indicating the tumor origin from squamous tissue [26].

#### 3.5.8. Pituitary Adenomas

In 2015 Wirth and colleagues described confocal reflectance and fluorescence characteristics of pituitary adenomas using a custom-built confocal benchtop microscope using DMB for staining. Tumor cells appeared uniformly round with an average diameter of about 15 µm, surrounded by bands of connective tissue. Reflectance image exhibited similar but lower contrast of the tumor cells [29]. Similar findings were reported by Charalampaki and Daali: pituitary adenomas appeared on CLE with topical AF as a mix of monomorphic cells characterized by dense, round to ovoid nuclei, with a higher cellular density if compared to normal brain tissue [18,33]. Bright cytoplasm was evident with CV staining in the work of Martirosyan and colleagues of 2018. Belykh and colleagues studied for the first time in 2020 the CLE characteristics of pituitary adenomas ex vivo using SF as contrast enhancer, demonstrating, among the characteristics listed above, heterogenous uptake of SF by cells creating a nuclear–cytoplasmic contrast, as well as contrast between neighboring cells [45].

#### 3.5.9. Central Neurocytoma

Sanai and colleagues reported confocal imaging characteristics of central neurocytoma [20]. Central neurocytomas usually presented uniform round cells organized in a honeycomb conformation, embedded against a background of arborized capillaries. Eschbacher in 2017 reported the confocal reflectance appearance of one central neurocytoma that showed sheets and clusters of cells with minimally pleomorphic dark, nonreflective nuclei and often abundant reflective cytoplasm. Such characteristics appeared similar to that of the corresponding H-E slides [32].

#### 3.5.10. Paraganglioma

Carotid body paraganglioma showed small nests of cells in a zellballen architecture, that may easily be appreciated on matched H-E slides in a work of Eschbacher [32].

#### 3.5.11. Hemangioblastoma

Hemangioblastomas are usually identified by large stromal cells mixed with a dense capillary network. Within such tumor, both cystic changes and areas of microhemorrhage may be appreciated, along with small vacuoles consistent with lipid. A rich vascular background is typically present [20,22].

#### 3.5.12. Metastases

As seen with reflectance and fluorescence imaging with MB (cell labeling) AF or AO, carcinoma nodules are typically surrounded by dense bands of fibrous tissue [38]. High cell density with a round-to-fusiform shape and big nucleus are a common feature in non-small cell lung carcinoma. Macrophages can be usually seen in fluorescence images and in corresponding pathological sections [18,28,29,33]. Looking at rectum carcinoma metastasis, glandular structures may be observed in adenocarcinoma specimens using confocal imaging [33]. Tissue examination of brain metastasis from mammalian carcinoma revealed a lawn-like, partly nester-like growth pattern between a focal distinctive, highly fibrotic stromal component in a work by Daali and colleagues [33].

#### 3.5.13. Epidermoid Tumors

Foersch and colleagues studied ex vivo application of CLE in brain tumors, analyzing, among others, one epidermoid tumor: crystal like structures were clearly seen as an indication of the epidermal origin, most likely representing desquamating epithelial cells. No nuclei could be found in the superficial cell layers [26]. Similar results were obtained by Charalampaki and colleagues in 2015, which underlined how, with AF staining, the granular layer appeared especially bright due to the high amount of stained nuclei by the dye [18]. CLE analysis showed neither calcification nor mitosis in the work of Daali in 2016 [33].

#### 3.5.14. Choroid Plexus Papilloma

Charalampaki and Daali reported confocal imaging characteristics of such tumor ex vivo [18,33]. Confocal images as well as traditional histology showed crypt structures with ordered, flat epithelial cells around fibrovascular cores. Cytological atypia, mitosis or necrosis was not detected.

#### 3.5.15. Plasmacytoma

In a study by Daali and colleagues such tumor appeared characterized by a high cell density, a high number of blood vessels and pleomorphism on CLE, after topical AF staining [33].

#### 3.5.16. Confocal Imaging in Non-CNS Tumors

Looking at other cerebral lesions, multinucleated giant cells may be identified in the case of sarcoidosis [11,15]. Two cavernomas and multiple abscesses were visualized under CLE imaging by Daali and colleagues. Cavernomas demonstrated to have high cell density, pleomorphism, high vascular malformations and spaces [33]. No information was given by the implementation of fluorescent dyes as reported by Martirosyan in 2018 [38]; abscesses typically showed a characteristic necrotic, purulent center, surrounded by a high cell density of mononuclear cells, astrocytes and other cell types [33]. Eschbacher in 2017 reported one case of fibromatosis observed under confocal reflectance microscopy without contrast administration. In the only available case of fibromatosis, the biopsy specimen showed sheets of cells with spindle features. The same authors reported the results from a hippocampectomy specimen resected for hippocampal sclerosis under confocal imaging: it appeared hypocellular and showed numerous processes suggestive of gliosis. Neuronal cell bodies were not visualized, although they were evident on the matched H-E slide [32].

### 3.6. The Clinical Role of Machine Learning in Confocal Imaging: A New Frontier for More Interpretable Images

Current CLE systems may obtain more than one image per second, hence, during a surgical operation, hundreds to thousands of images may be collected and stored. Such number of data may become rapidly overwhelming for neurosurgeons and neuropathologists when a quick selection of diagnostic or significant image is necessary, especially if novel CLE systems are being used on the fly in real time, intraoperatively. Thus, overcoming such barriers would be a key factor in rendering confocal imaging a feasible and practical technology for the neurosurgical operating room. Moreover, beyond a “quantity” reason, manually filtering out nondiagnostic images is challenging also due to the novel and often unfamiliar appearance of tissue morphology if compared to histology and due to the great variability among images from the same tumor type and potential similarity between images from other tumor types. To date, despite its promising diagnostic potential, interpreting the gray tone fluorescence images of last available systems can be difficult for untrained users.

As performed in other subspecialties, the applications of machine learning in medical imaging have greatly increased in the last years, resulting in numerous computer-aided detection and diagnosis systems [54]. Regarding this aspect, the application of deep learning models for automatic detection of the diagnostic CLE images has been suggested and appears promising. Applications of machine learning for confocal imaging in neurosurgery have been performed still by few researchers. Entropy-based filtering is one of the simplest ways to filter out non-diagnostic CLE images. In a study by Kamen et al., CLE images obtained from brain tumors were classified automatically through an entropy-based approach, with the aim of removing non informative images from databases [55]. In this work meningioma and glioma were differentiated using bag of words and other sparse coding methods. However, some authors state that entropy might not be an ideal method since many obtained images have nearly as high entropy as diagnostic ones [54].

The group of Izadyyazdanabadi and colleagues recently developed an ensemble of deep convolutional neural networks that can automatically evaluate the diagnostic value of CLE frames within milliseconds. In the first work the authors implemented AlexNet, a deep-learning architecture, that was used in a 4-fold cross validation manner analyzing 16,795 images from 74 CLE-aided brain tumor resections. Average model accuracy on test data was 91% overall, suggesting that a deeply trained AlexNet network can achieve a model that reliably and quickly recognizes diagnostic CLE images. In a subsequent work the model was refined and upgraded, increasing the number of images analyzed, training the model in multiple regimes and accomplishing an interobserver study [54]. To further improve the diagnostic quality of CLE images, the same group in 2019 tested an image style transfer, a neural network method for integrating the content and style of two images. This was done through minimizing the deviation of the target image from both the content (CLE) and style (H-E) images. Such style transferred images were then compared to conventional H-E histology by neurosurgeons and a neuropathologist who validated the quality enhancement. With this method the authors were able to provide images more easily interpretable than the original CLE images, allowing a real-time, cellular-level tissue examination using CLE technology more similar to the conventional appearance of H-E staining [56].

Apart from these applications, another aspect that could be approached using machine learning models is the issue of securing the diagnostic images obtained. One important issue is to address how to share sensitive data while limiting disclosures and limiting their sharing by ensuring the sufficient data utility to all involved users. For instance, a limited restriction of data access may lead to a decrease in information content too, that might affect the diagnostic potential of images [57,58]. Kaissis and colleagues in a 2020 work stated that the widespread adoption of secure and private artificial intelligence solutions will require targeted multi-disciplinary research and investigations in many fields with multiple objectives such as: (1) to decentralize data storage and federated learning systems, replacing the actual paradigm of data sharing and centralized storage; (2) to counteract the drawbacks of the individual techniques already available (neural network operations, functional encryption); (3) to increase cryptographic expertise [58]. This is a very hot topic in current medical literature that has not been deeply touched and described when looking at confocal imaging systems in neurosurgery. Hence, further works in this field would be welcomed and are awaited.

The amount of data available and necessary for analysis has already eclipsed human capabilities and a physician will not have time to inspect thousands of images per single case, especially if considering the on-the-fly use that modern CLE systems permit. Therefore, a theragnostic approach must be employed as we step toward of ever-increasing information in neurosurgery in search of personalization. As a consequence, further studies in this context are awaited and welcomed.

## 4. Summary

While other authors brilliantly reviewed intraoperative imaging modalities for neurosurgical practice as well as preclinical applications of confocal imaging in neurosurgery, this is the first review specifically focused on actual clinical applications of confocal imaging in neurosurgery [23,59,60]. Main objective of this review was to elucidate which role, by now, confocal imaging technology may have in neurosurgical operations in obtaining valuable in vivo or ex vivo optical biopsies, possibly guiding intraoperative differential diagnosis and improving extent of resection.

The principle that resides beyond confocal microscopy technology dates back to the 1950s [18]. In contrast to conventional light microscopy, in which the entire specimen is illuminated, the main principle of confocal microscopy is that, in the beam path of the detected light, a pinhole is positioned to block the light coming out of the focal plane. This leads to a reduction in the depth of field, which in turn results in the improvement of the resolution along the optical axis (z plan). This method has been used in in biology for years, but only recently these devices have landed to clinical therapy to characterize cells intraoperatively. The use of confocal microscopy has also been coupled with the use of fluorescent dyes to achieve rapid histopathological diagnosis in numerous fields, including gastroenterology, gynecology, dermatology and ophthalmology [12,13]. More recently, CLE established itself in the field of gastroenterology as a miniaturization and evolution of the confocal microscopy method and, recently, this technology landed to neurosurgical fields. Starting from the experience gained in other fields, in recent years different authors studied confocal imaging technology in neurosurgery, trying to identify its possible role in intraoperatively detecting and characterizing tumor cells, providing a possible immediate diagnosis (similarly to a frozen section), providing delineation of borders between tumor and normal tissue on a cellular level, hopefully making surgical margins more accurate than ever before. Different fluorescent dyes have been used in neurosurgical clinical practice, in both ex vivo and in vivo experiences. While in in vivo experiences SF and 5-ALA resulted to be the most used fluorescent dyes, in ex vivo imaging an expanded repertoire of fluorophores on brain tissue, many of which are considered too toxic for in vivo use, could be used.

To date, analyzing actual pertinent literature we were able to find just seven clinical in vivo studies and twenty-one ex vivo experiences that matched our strict inclusion/exclusion criteria. Comprehensively analyzing these studies, we feel there is still insufficient data to make any definitive conclusion on the real usefulness of confocal imaging technology in improving intraoperative tumor diagnosis and eventually extent of resection in neurosurgery. One of the reasons for this is the paucity of studies, to date, that rigorously analyzed this technology. For instance, many studies retrospectively analyzed confocal imaging from a purely qualitative point of view, reporting hallmarks of different pathologic tissues, as seen with or without the aid of fluorescent dyes [27,30,31,36,37]. This aspect would be sufficient to check the “histopathologic” potential of this technology but would be completely insufficient if the possibility of using this technology to obtain intraoperative real time diagnosis is investigated. This aspect would be possible only with blinded prospective studies. Moreover, many studies were carried out using laboratory confocal microscopes (with specimens sent from OR to laboratory as soon as collected during surgery) or confocal machines not specifically designed for neurosurgical purposes [31,34,35,38]. Thus, the real possibility of integrating this technology in concrete clinical practice is still questioned. Looking at this specific purpose (understanding the real adjunct that this technique may carry in routine neuro-oncological practice), very often no data on sensibility and specificity of the different studies are reported, nor the “blindness grade” of the experience, in respect of which information are available to the confocal imaging reader at the moment of diagnosis. Thus, such aspect was voluntarily underlined in Table 1 and Table 2 for each study.

Reviewing actual literature, looking at the real significance that confocal imaging may have in neurosurgery, we feel that two points deserve deep discussion:
Confocal imaging may be used similarly to a frozen section analysis: to obtain an intraoperative differential diagnosis. Looking at this specific purpose, scarce but pertinent results are available in neurosurgical literature, although data should be considered and interpreted in light of two aspects: the “blindness grade” of each study and the place where the hypothetical diagnosis was made (i.e., in OR or in the lab or in office), to check for its possible role to be implemented in routine clinical practice.For instance, in the work of Snuderl the bioptic samples were transferred to another institute to be studied on a benchtop microscope and, although the design of the study was built to simulate an intraoperative frozen section evaluation process, testing sensitivity and specificity, the real design of the study was far from being possibly integrated in a neurosurgical OR [28]. Foersch in 2012 used Optiscan (now dismissed) to make diagnosis using a benchtop confocal microscope [26]. Additionally, in this case, although raters were blinded to histology and macroscopic appearance of the tumor, diagnosis was not given in the OR, during neurosurgical procedure. Both Daali and Breuskin reported interesting results regarding sensibility/specificity of this technology in obtaining intraoperative diagnosis, with an acceptable grade of blindness of pathologists. No sensibility/specificity data were given in both studies regarding morphological characterization of different tumors [19,33]. Similarly, no morphological quantitative characterization was carried out by Eschbacher in 2017, where pathologists were aware of eventual contrast enhancement and location of tumors analyzed (lower grade of blindness) [32]. In this context, the work of Belykh in 2020 should be considered as a miliary stone, due to the extremely rigorous blinded study in analyzing CLE images from both neuropathologist and experienced/unexperienced neurosurgeons’ points of view and the huge amount of tumors investigated [41]. Looking at our work, in 2020 we reported the results of a rigorous blinded and prospective study where an optimal diagnostic concordance was obtained at the central core of 15 HGGs analyzed [42]. Even though one of the major limits of this study was the fact that just one hystotipe of tumor was analyzed, hence rendering it unsuitable to study the differential diagnosis ability of Convivo (programmed for the next in vivo study), we reported an optimal concordance among CLE imaging and histopathological/frozen section analysis at the central core of the different cases, with a rigorous blinded design. Moreover, we performed for the first time a different analysis for specimens taken at central core or at tumor margins (see the second point of this discussion). In conclusion, still more data on the real possibility of obtaining prospectively and blindly an intraoperative differential diagnosis using confocal imaging are necessary and awaited.Confocal imaging may be used to check for the presence of tumor at the tumor margins, possibly increasing EOR. As stated before, from a methodological point of view, a real sensibility and specificity calculation would be possible only for pure in vivo experiences, given the need of optical biopsies on healthy brain parenchyma (negatives, see above). To date, the only solid statistic study that prospectively analyzed in vivo sensibility and specificity of CLE imaging obtained at both tumor central core and margins was the one of Martirosyan in 2016 [11]. Nevertheless, in this study no specific statistic differentiation was made among specimens taken at central core and at transition zones. Hence, a real ability in identifying tumor tissue at borders was not investigated. In addition, considering ex vivo studies, a very paucity of data regarding biopsies taken at tumor margins is present. As anticipated, our work in 2020 was the first prospective and blinded ex vivo study to specifically check for ability of Convivo system to identify tumor tissue at borders in HGGs [42]. Comparing Convivo and frozen section analyses, we obtained a high rate of concordance at tumor borders in both obtaining a diagnosis and categorizing morphological patterns. Unfortunately, we were not able to extend this study also to meningiomas, metastasis and other tumor types, reason for which a prospective, blinded in vivo study using Convivo analyzing different CNS tumors at both central core and borders is ongoing in our Institute. Considering the paucity of data regarding this specific issue, further studies in this field are awaited to discover how sensitive and specific such systems may be when used to check for tumor tissue at borders.

Another aspect to be considered is that, similarly to other operator-dependent technology, there is a learning curve in using confocal systems. The probe position in fact may be changed multiple times during surgery to acquire optimal images from a given area. Often, it is not feasible to image the entire resection cavity due to the small field of view of the CLE although there is no practical necessity to screen the entire surgical bed, given that the rationale of using this technology is to acquire “optical biopsies” in selected regions. Moreover, the process of interpretation of the black, gray and white digital images compared with inspection of standard H-E-stained tissue preparations requires experience. Few authors studied this aspect as an ancillary aspect of their work, but to date no clear data are available on quantification and strategies to decrement the learning curve associated with this technology.

In conclusion, considering pertinent literature, it is reasonable to suppose that further studies may elucidate whether subsequent routine use of CLE-assisted surgery may significantly improve both the diagnosis and the treatment of tumorous processes, offering patients an increased survival rate and quality of life.

## Figures and Tables

**Figure 1 jcm-10-02035-f001:**
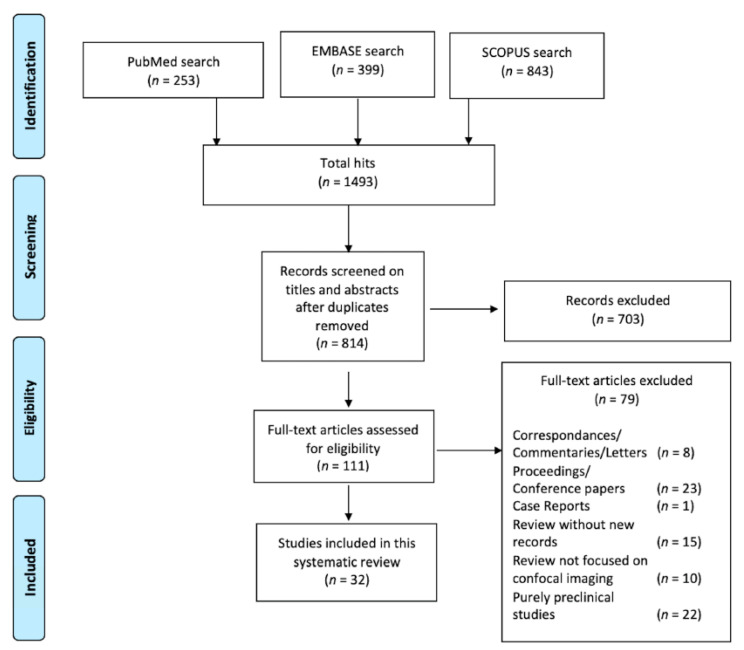
The flowchart of search hits and the different Preferred Reporting Items for Systematic reviews and Meta-analyses (PRISMA)-guideline selection phases, from the initial search and the follow-up search (b), resulting in the total amount of 32 included articles.

**Figure 2 jcm-10-02035-f002:**
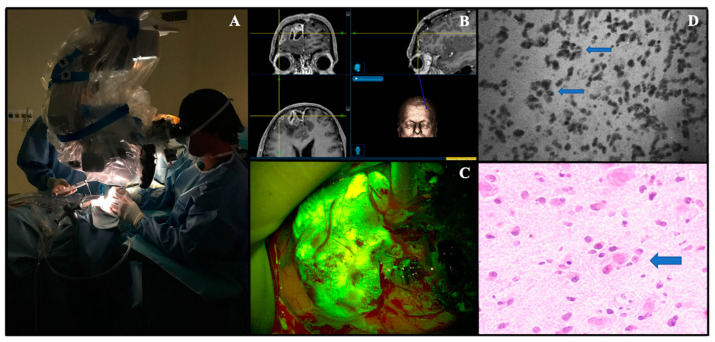
In vivo Convivo case (Besta Neurological Institute, Milan, Italy). (**A**). the confocal probe is dressed with its appropriate sterile sheath and used directly upon cerebral surface. (**B**). Preoperative magnetic resonance with contrast administration images loaded on the neuronavigation system (Stealth S8-Medtronic) of a right frontal parasagittal anaplastic oligodendroglioma, IDH mutant (WHO grade III). (**C**). Intraoperative view of fluorescein-guided removal of the tumor under YELLOW560 filter activation on Pentero microscope (Carl Zeiss Meditec). (**D**)**.** Convivo in vivo image taken at the center of the tumor showing tumor cells along with typical perineural satellitosis (small arrows), that can be easily found on relative histopathological image as well (H-E, big arrow, (**E**)).

**Figure 3 jcm-10-02035-f003:**
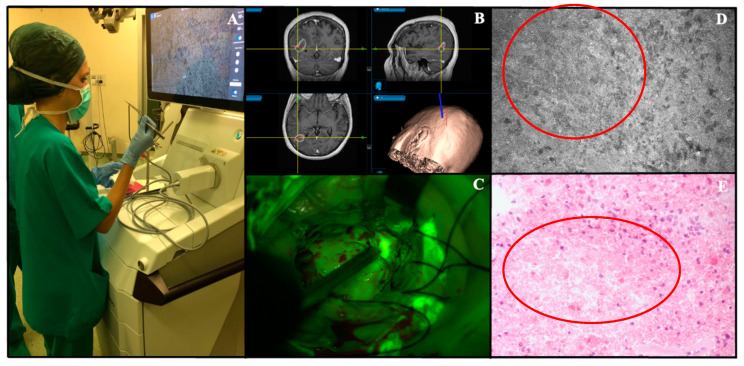
Ex vivo Convivo case (Besta Neurological Institute, Milan, Italy). (**A**). The Convivo station is placed inside OR and confocal imaging analysis is performed ex vivo during surgical operation. (**B**). Preoperative magnetic resonance with contrast administration images loaded on the neuronavigation system (Stealth S8-Medtronic) of a right parieto-occipital GBM. (**C**). Intraoperative view of fluorescein-guided removal of the tumor under YELLOW560 filter activation on Pentero micro-scope (Carl Zeiss Meditec). (**D**). Convivo ex vivo image taken at the center of the tumor showing sparse tumor cells among focal necrosis foci, characterized in CLE images as areas of low cellular density (red circle); the same morphological characteristic may be appreciated in corresponding histopathological H-E image (red circle, (**E**)).

**Table 1 jcm-10-02035-t001:** In vivo experiences (7 works). Legend to the table: CLE: confocal laser endomicroscopy; HGGs: high-grade gliomas; iv: intravenously; LGG: low-grade gliomas; SF: sodium fluorescein; 5-ALA: 5-aminolevulinic acid.

Study, Year	Confocal System Used	Fluorophore Used (Dosages, Protocol)	*N*. of pts	Pathologies Treated	Main Findings	Sensibility/Specificity; Diagnostic Accuracy	Blinding Level of the Study
Sanai et al. (2011) [20]	Optiscan 5.1	SF (5 mL, 10% in saline solution, injection immediately before imaging)	35	13 LGGs 8 HGGs 8 Meningiomas 3 radiation necrosis	First in vivo experience in humans with confocal laser technology, using SF as contrast enhancer, demonstrating feasibility of in vivo confocal imaging in neurosurgery	/	/
Sanai et al. (2011) [21]	Optiscan 5.1	5-ALA (20 mg/kg, 3 h before surgery)	10	10 LGGs	First in vivo experience with 5-ALA as contrast enhancer: CLE may be used in conjunction with 5-ALA to detect LGGs at borders	/	/
Eschbacher et al. (2012) [22]	Optiscan 5.1	SF (25 mg iv, 2–5 min before CLE imaging)	50	24 Meningiomas 12 HGGs 8 LGGs 4 Schwannomas 2 Other tumors	First report of CLE accuracy in obtaining in vivo diagnosis	ACCURACY (ex vivo analysis): 92.9% in obtaining correct diagnosis (26/28)	Blindness of pathologist regarding type of tumor operated, available information on location and radiological enhancement; possibility to choose diagnosis among a list of possible tumors
Martirosyan et al. (2016) [11]	Optiscan 5.1	SF (5 mL, 10% in saline solution, iv injection 5 min before imaging)	74	30 Meningiomas 14 Other tumors 13 HGGs 7 No tumors 4 Schwannomas 1 Metastasis	First report of operative data about CLE imaging in vivo; first report of sensibility and specificity for in vivo imaging of HGGs and meningiomas	SENSIBILITY/SPECIFICITY for:→HGGs: 91%/94% →Meningiomas: 97%/93%	Blindness of the study not specifically defined (“Images were reviewed by a neuropathologist and 2 neurosurgeons who were not involved in the surgeries”)
Pavlov et al. (2016) [17]	Cellvizio	5-ALA (4 h before surgery; SF 500 mg/5 mL)	18	6 HGGs 2 LGGs 1 Lymphoma	Feasibility of CLE with 5-ALA and SF; first application of CLE to in vivo brain stereotactic biopsy	/	/
Charalampaki et al. (2019) [23]	Multispectral fluorecence microscope + Cellvizio	ICG (50 mg iv 1 h before tumor removal)	13	5 Gliomas 3 Meningiomas 3 Metastases 2 Schwannomas	Feasibility of in vivo concomitant use of multispectral surgical microscope with Glow800 software and CLE imaging in brain tumor surgery; feasibility of inserting CLE probe in endoscopic channel for looking “behind the corner” in brain tumor surgery	/	/
Hohne et al. (2021) [24]	Convivo	SF (5 mg/kg, 10% in saline solution, injected intraoperatively (various times)	12	5 Metastasis 4 HGGs 2 LGGs 1 Gliosis/recurrent hemangiopericytoma	Feasibility of in vivo Convivo system in humans in different kind of brain tumors	/	/

**Table 2 jcm-10-02035-t002:** Ex vivo experiences (21 works). Legend to the table: AO: acridine orange; CLE: confocal laser endomicroscopy; CV: cresyl violet; DMN: demeclocycline; HGGs: high-grade gliomas; ICG: indocyanine green; iv: intravenously; LGG: low-grade gliomas; MB: methylene blue; OR: operating room; PpIX: protoporphyrin IX; SF: sodium fluorescein; 5-ALA: 5-aminolevulinic acid.

Study, Year	Confocal System Used	Fluorophore Used (Dosages, Protocol)	*N*. of pts	Pathologies Treated	Main Findings	Sensibility/Specificity; Diagnostic Accuracy	Blinding Level of the Study
Schlosser et al. (2009) [25]	Optiscan	AF (0.05%, topical administration)	12	9 HGGs 3 Meningiomas	Pilot study demonstrating the feasibility of CLE imaging ex vivo in brain tumor surgery	/	/
Foersch et al. (2012) [26]	Optiscan	AF (50 microliters topical or SF 50 microliters topical)	15	6 Meningiomas 3 Other tumors 2 Schwannomas 2 Healthy brain specimens 1 HGG 1 Metastasis	First attempt to calculate and report a diagnostic accuracy for ex vivo CLE imaging with AF	ACCURACY in diagnosis on 35 preselected images from 7 different tissues→87% for non-clinicians →93% for clinical experts	5 raters blinded to histology and macroscopic appearance of tumors
Wirth et al. (2012) [27]	Multimodal confocal microscope	MB (1% topical around 2–5 min before imaging)	119	41 HGGs 25 Metastases 14 Meningiomas 11 LGGs	Feasibility of multimodal confocal reflectance and fluorescence imaging for histologic assessment of brain tumors ex vivo	/	/
Snuderl et al. (2012) [28]	Multimodal confocal microscope	MB (1% topical around 2–5 min before imaging)	37	10 Metastases 9 HGGs 8 Non tumors 3 LGGs	Sensibility/specificity available for ex vivo imaging with a multimodal confocal reflectance/fluorescence imaging	SENSIBILITY/SPECIFICITY: →in identifying normal vs abnormal tissue of 95%/100% →in identifying glial vs non-glial tumor of 83%/90% →in making final correct diagnosis (neuropathologists) of 88%/100%	13 pathologists (9 generalists and 4 neuropathologists), all without previous exposure to confocal imaging technique, assessed each case answering to such questions: →is the tissue normal brain or abnormal; →if abnormal, is it a glial or nonglial neoplasm; →if glial, is it a LGG or a HGG; →if nonglial, is it meningioma or metastatic carcinoma.
Breuskin et al. (2013) [30]	EndoMag1	MB (topical around 20 min before imaging)	>50	Not further described	First feasibility study on red light CLE in brain tumors surgery	/	/
Georges et al. (2014) [31]	Benchtop confocal microscope	Only reflectance microscopy	2	1 HGG 1 Radiation necrosis	CLE proposed as a feasible method to distinguish among tumor and radiation necrosis prior to specimens biobanking	/	/
Wirth et al. (2015) [29]	Multimodal confocal microscope	DMN (0.75 mg/mL, staining for 20 min 1–2 h after surgery)	14	7 HGGs 4 Metastases 2 Meningiomas 1 Pituitary adenoma	Feasibility study of DMN used as optical contrast enhancer for HGG tumor cells; morphological characteristics of different CNS tumors are presented	/	/
Forest et al. (2015) [34]	VivaScope 2500	AO (undefined dosage)	19	9 Meningiomas 7 HGGs 3 Metastases	Feasibility study of the implementation of a dermatological confocal microscope on a routine use for the most frequent brain tumors. First ex vivo human study to implement AO as contrast enhancer.	/	/
Charalampaki et al. (2015) [18]	Cellvizio	AF (0.05% topical)	150	47 HGGs 32 Metastases 30 Meningiomas 16 Other tumors 13 Schwannomas 12 LGGs	Descriptive study showing CLE features of different CNS tumors	/	To establish accuracy and interobserver agreement a set of confocal images (n = 100) of 20 different tissues were selected and presented to 2 groups of raters: nonclinical and clinical experts, blinded to the macroscopic appearance and the histopathological diagnosis of routine pathology.
Daali et al. (2016) [33]	Cellvizio	AF (0.01 mg/mL topical)	258	74 Meningiomas 69 Metastases 50 Other tumors 47 HGGs 7 LGGs	Prospective descriptive study reporting the overall accuracy in making diagnosis for ex vivo AF CLE imaging	ACCURACY in obtaining diagnosis of 89%, calculated on preselected images from 258 cases	Images were evaluated by 4 different evaluators (surgeons and neuropathologists). The traditional histopathological findings were blinded to both groups.
Breuskin et al. (2017) [19]	EndoMag1	Only reflectance microscopy	100	34 Meningiomas 32 HGGs 16 Metastases 10 LGGs 8 Schwannomas	First study assessing sensibility and specificity for identifying brain tumors ex vivo with red light confocal imaging without prior contrast administration	SENSIBILITY/SPECIFICITY for diagnosis of: →HGGs 85%/81% →Meningiomas 82%/95% →LGGs 90/93% →Schwannomas 87%/100% →Metastases 7%/94%	The CLE investigator was blinded for patient data and for results of instantaneous sections.
Eschbacher et al. (2017) [32]	Benchtop confocal microscope	Only reflectance microscopy	76	25 Meningiomas 24 Other tumors 10 Pituitary adenomas 8 HGGs 7 LGGs 5 Normal pituitary glands 4 Schwannomas 4 Metastases 3 Treatment effect	Excellent image quality study, also reporting a blinded interpretation of acquired images by neuropathologists and general pathologists	ACCURACY: in→asserting tumors vs non tumor: 91.5 % for general pathologists 97.9 % for neuropathologists →labeling lesions with corrected diagnosis: 85.1% for general pathologist 95.8% for neuropathologist	Preselected 47 images, analyzed blindly by neuro and general pathologists without prior experience in confocal laser imaging. Pathologists were aware of eventual contrast enhancement and location of tumors.
Wei et al. (2017) [35]	Benchtop confocal laser microscope	5-ALA (20 mg/kg before surgery per os)	14	14 HGGs and LGGs	Feasibility of ex vivo confocal microscopy analysis after PpIX administration	/	/
Yoneyama et al. (2017) [36]	Benchtop confocal laser microscope	5-ALA (20 mg/kg before surgery per os)	More than 20	More than 20 HGGs	Fluorescence intensity and bright-spot analysis using 5-ALA as contrast enhancer may help in distinguishing a tumor region, differentiating between infiltrating tumor and normal regions.	/	/
Martirosyan et al. (2018) [38]	Benchtop confocal laser microscope	AF (0.05%) AO (0.01%) CV (0.02%) MB (82%) ICG (0.6%)	106	32 Other tumors 30 Meningiomas 19 Gliomas 13 Pituitary adenomas 9 Metastases 3 Non tumor	First study to implement different fluorescent dyes for ex vivo imaging, setting procedural “standards” and reporting high image quality, especially for AF and AO staining. AF and AO staining resulted to be the best option for the great majority of tumors investigated	ACCURACY: Correct identification of: ➔86.7% meningiomas (also correct subtype) ➔89% gliomas (21% correct subtyping) ➔100% pituitary adenomas ➔55.6% metastases	/
Mooney et al. (2018) [39]	Benchtop confocal laser microscope	Only reflectance microscopy	11	11 Pituitary tumors	Feasibility study for confocal reflectance microscopy without contrast enhancers for pituitary adenomas, reporting also accuracy in their identification	ACCURACY in making proper diagnosis on 16 preselected images: 94%	16 representative confocal images from the 11 cases were selected by the neuropathologist (7 images of normal adenohypophysis and 9 images of pituitary adenoma), then presented in a blinded fashion to a second dedicated neuropathologist who had no prior knowledge of the cases.
Belykh et al. (2018) [40]	Convivo	SF (2–5 mg/kg iv 5–60 min before imaging) AO (0.1%) AF (0.1%)	31	11 Other tumors 9 HGGs 4 Metastases 3 Schwannomas 3 Meningiomas 1 LGG	Feasibility study for Convivo ex vivo analysis of brain tumors.	/	/
Yoneyama et al. (2019) [37]	Benchtop confocal laser microscope	5-ALA (20 mg/kg before surgery per os)	> 9	6 GBMs characterized by 5-ALA induced fluorescence 3GBMs characterized by no 5-ALA induced fluorescence 5 additional specimens (2 HGGs, 1 LGG, 1 recurrent GBM, 1 nerve sarcoma, 1 normal tissue)	Bright-spot analysis may be of help in distinguishing tumorous vs non tumorous tissue also in GBM without 5-ALA induced fluorescence and in other tumor subtypes	/	/
Belykh et al. (2020) [45]	Convivo	SF (2 mg/Kg iv before biopsy collection; optimal timing reported to be from 1 min to 10 min before biopsy)	9	9 Pituitary adenomas (13 biopsies)	Feasibility of portable Convivo probe implementation in cadaver heads through a trans-sphenoidal corridor for pituitary adenomas; ex-vivo study of accuracy in their identification with SF as contrast enhancer.	ACCURACY in diagnosing pituitary adenomas with frozen sections as standard: →“definitively” for 13/16 specimens. →“favoring” for 3/16 specimens.	A neuropathologist with experience interpreting CLE images, but who was not involved in the surgical procedures, reviewed the CLE digital images as well as frozen and permanent section slides.
Belykh et al. (2020) [41]	Convivo	SF (2 to 5 mg/kg iv upon induction of anesthesia, 5 mg/kg during surgery for CLE contrast improvement)	47	29 HGGs 7 Meningiomas 4 Metastasis 3 LGGs 1 Choroid plexus carcinoma 1 Cranyopharingioma 1 Schwannoma 1 Arterovenous malformation	Very detailed quantitative and descriptive analysis of different brain tumors, along with autofluorescent cells characteristics classification; first time where a second SF injections was used to improve diagnostic power of CLE; blinded study in analyzing CLE images from both neuropathologist and experienced/unexperienced neurosurgeons (regarding CLE imaging interpretation)	DIAGNOSTIC ACCURACY: positive predictive value of: 98% for gliomas 91% for meningiomas 83% for metastasis with an overall diagnostic accuracy of 75% for blinded neuropathologist. DIAGNOSTIC ACCURACY for all biopsies blinded for analysis by a neurosurgeon with experience interpreting CLE images was 78% vs 71% for the neurosurgeon without CLE image-reading experience.	→The neuropathologist had no clinical information except that the biopsy had been performed during an intracranial procedure →Two neurosurgeons (experienced and unexperienced) reviewed a set of CLE images, after being instructed on the key histologic features on CLE images. No information was provided to the neurosurgeons or general pathologist regarding case history, imaging or diagnosis
Acerbi et al. (2020) [42]	Convivo	SF (5 mg/kg at anesthesia induction)	15	15 HGGs	First available study where the ability of Convivo in obtaining intraoperative diagnosis and categorizing morphological patterns at both central core and tumor margins was assessed prospectively and based on a near real-time, blinded interpretation of the pathologist during surgery in OR.	ACCURACY in: →Obtaining a diagnosis (compared to frozen sections): 80% at central core, 80% at tumor border →Categorizing patterns (compared to frozen sections): 93.3% at central core and 80% at tumor margins →Obtaining a diagnosis (compared to permanent sections): 80% at central core, 67% at tumor border →Categorizing patterns (compared to permanent sections): 86% at central core and 67% at tumor margins.	A dedicated pathologist was asked to judge in near real-time intraoperatively if the tissue represented tumor tissue, to provide a possible intraoperative tumor diagnosis, and to categorize eventual morphological patterns (blinded to frozen and permanent section results)

## Data Availability

The data presented in this study are available in the present article. Further data on review search or data regarding clinical cases are available upon request to the corresponding Author (F.A.).

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
