# Peer review of "Confocal Laser Microscopy in Neurosurgery: State of the Art of Actual Clinical Applications"

_jcm, 2021, doi:10.3390/jcm10092035_

Round 1
Reviewer 1 Report
The text is well written. The chronological development of the CLE is very well reflected. The indication for the use of the technology is analyzed in detail. I think you should explain the use of artificial intelligence and its role in securing diagnostics.
Reviewer 2 Report
Dear Authors,
Congratulations to that very well researched and well written review about confocal laser microscopy in neurosurgery.
Material and Method were well structured and described. You include a comprehensive and sufficient collection of articles in this issue. However, I was missing clear information’s about the total numbers in the text, of both: in vivo experiences and ex vivo experience. The particular papers and total numbers of papers should be specify in more detail.
For Example page 24…we were able to find just few clinical in vivo studies... please specify and revise in the whole manuscript.
For a better understanding, please insert the total numbers of papers taken into account.
Tumor classification and splitting are understandable.
In Summary, your review encouraged others to move forward in this special issue. It offers a promising basis to investigate more and other aspects about the possibilities of confocal imaging technology in neurosurgical operations.
